# PVTree: A Sequential Pattern Mining Method for Alignment Independent Phylogeny Reconstruction

**DOI:** 10.3390/genes10020073

**Published:** 2019-01-22

**Authors:** Yongyong Kang, Xiaofei Yang, Jiadong Lin, Kai Ye

**Affiliations:** School of Electronic and Information Engineering, Xi’an Jiaotong University, Xi’an 710049, China; kangyong@stu.xjtu.edu.cn (Y.K.); xfyang@xjtu.edu.cn (X.Y.); jiadong66@stu.xjtu.edu.cn (J.L.)

**Keywords:** phylogenetic tree, sequential pattern mining, alignment free, multiple sequence alignment

## Abstract

Phylogenetic tree is essential to understand evolution and it is usually constructed through multiple sequence alignment, which suffers from heavy computational burdens and requires sophisticated parameter tuning. Recently, alignment free methods based on k-mer profiles or common substrings provide alternative ways to construct phylogenetic trees. However, most of these methods ignore the global similarities between sequences or some specific valuable features, e.g., frequent patterns overall datasets. To make further improvement, we propose an alignment free algorithm based on sequential pattern mining, where each sequence is converted into a binary representation of sequential patterns among sequences. The phylogenetic tree is further constructed via clustering distance matrix which is calculated from pattern vectors. To increase accuracy for highly divergent sequences, we consider pattern weight and filtering redundancy sub-patterns. Both simulated and real data demonstrates our method outperform other alignment free methods, especially for large sequence set with low similarity.

## 1. Introduction

Construction of a phylogenetic tree is one of the fundamentals in bioinformatics. It describes how a protein (gene) family might have been evolved. In the past decades, pairwise and multiple sequence alignment (MSA) [1,2,3] with maximum likelihood function are the trending methods used to build an accurate phylogenetic tree [4,5]. However, alignment-based methods inherit five pitfalls. First, it is a time consuming process to align hundreds or thousands of sequences [6]. Second, the alignment results depend on various parameters such as gap opening and extension penalties and it might affect the final phylogeny [7]. Third, these methods will be affected by the guild trees during alignment [8]. Fourth, sequence distance emphasizes too much on the well-aligned regions while ignore small fragments with valuable information to estimate phylogeny. Last but not least, sequences become much more divergent as substitutions and indels accumulating thus makes it even harder to acquire qualified alignment [9]. During species evolution, though sequences of species from the same origin divergent differently under selection pressure, some fragments still conserved between species. For example, homolog proteins that result from ancient whole genome duplications under high mutation rate, robust cysteine-rich proteins that are highly tolerant of mutation. Therefore, it is important to analyze these low similarity sequences.

Having all aforementioned issues in mind, efficient and accurate methods for phylogeny reconstruction are demanding. Recently, lots of alignment free methods are proposed to address the challenges from next-generation sequencing data [10]. Besides some approaches based on graphical representation and information theory, alignment free methods mainly include two types: first, k-mer based methods like feature frequency profile (FFP) [11] and CVTree (CV) [12]. They use a sliding window with a predefined length to extract features in sequence sets and convert each sequence into a feature frequency vector, which is used to compute the distance matrix. Phylogenetic tree can be further constructed by Neighbor joining [13] or UPGMA [14]. Different from above mentioned methods that using exact word match, spaced word (SW) [15] discovers patterns with fuzzy matches and skipped positions, that is spaced-k-mer. But for large divergent sequence set, fixed feature length will result in redundant sequences features and thus greatly increase computational expenses. Second, substring based methods, such as average common substring (ACS) [16] and underlying approach (UA) [17]. They extract common substrings of each sequence pair for distance matrix calculation. However, these method just consider the similarity or distance of pairwise sequence while ignore the potential relationship of global sequences and thus lost differential information.

To overcome mentioned issues, we propose a sequential pattern mining based method to construct phylogenetic tree, which is especially pronounced for highly divergent sequences. As shown in Figure 1, we first extract frequent sequential patterns with position information and measure the distance among all sequences based on the extracted patterns. Using the feature space formed by these patterns, the sequence sets can be converted into a pattern vector sets. Then a phylogenetic tree can be built based on these vectors. Our method cannot only mine exact match patterns but also fuzzy matching, such as fuzzy pattern AxT contains a wildcard x [18]. Besides fuzzy matching, we add weight to each pattern and filter redundant patterns to achieve a high-quality feature space. Experiments on simulated and real data, our method is proved to be reliable and outperformed towards other methods.

## 2. Materials and Methods

### 2.1. Simulated and Real Data

We generated simulated protein sequences via Rose [19], which implements a probabilistic model of evolution to generate protein-like sequences. Using a simulated evolutionary tree generated by Rose as a guide, a set of related sequence families is created from a common ancestor sequence by inserting, deleting, and replacing characters. In this simulation, the “true” evolution history and “correct” multiple sequence alignments are recorded for further algorithm evaluation. The average evolutionary distance between generated sequences is determined by the relatedness parameter which is associated with point accepted mutation (PAM).

We use BAliBASE3.0 [20] as the real data benchmark, a widely used and specifically designed sequence alignment dataset. Particularly, the reference alignments result provided based on 3D structural superposition and manually refined. In addition, we construct a phylogenetic tree as the evolution reference by using the Maximum Likelihood method [21] in MEGA7 [22] based on BALIBASE sequence alignments since reference tree is not provided by BALIBASE.

### 2.2. Sequential Pattern Mining from Unaligned Sequences

In this study, we adapt our previous method [18] on unaligned protein sequences for frequent pattern discovery. Briefly, a pattern is the ordered combination of items from alphabet A associated with wildcard character x. Specifically, in our study the alphabet indicates amino acids or nucleotides. Support of a pattern is defined as the ratio of sequences that contains the pattern. In order to build appropriate pattern vectors, each pattern is discovered with corresponding position information (summarized in Algorithm 1).

For a given set of input sequences, the mining process begins with an empty pattern Λ and its initial projected database ($S_{\Lambda }$). A pattern’s projected database contains all sequences start with current pattern, and we mark the occurrence of last item within a pattern. The pattern a grows by item that satisfy the support threshold (b or x in Algorithm 1) and the projected database ($S_{a}$) that marks all occurrence of the new appended item (b or x) is updated at each growth iteration. The support of a is denoted as |$S_{a}$|.

**Algorithm 1** Mining Sequential Pattern (Note: *min_support* is the minimum support threshold, *min_mon_wc* is the minimal non-wildcard threshold, and last (*a*) is the last amino acid or nucleotides)1: Mining (a,$S_{a}$)2:  **if** |$S_{a}$| ≥ min_support
**then**3:    **if**
non_wc (a) ≥ min_mon_wc
**and** last (a) ≠ x
**then**4:     **if**
a is a closed pattern **then**5:       **report**
a, |$S_{a}$|6:    **for** each residue b or x
**do**7:      $a^{\prime }$: = a with b or x appended to it8:      Mining ($a^{\prime }$,${S_{a}}^{\prime }$)

Since sub-patterns of frequent patterns are also frequent [23]. Therefore, we just save closed patterns to remove redundancy during pattern mining. The closed patterns X are defined as that there exists no proper super-pattern Y such that Y has the same support as X in sequence set, that satisfy two constraints: the support larger than or equal to the minimum support (min_support); the number of non-wildcard items larger than or equal to minimal non-wildcard (min_non_wc).

### 2.3. Converting a Sequence to a Weighted and Non-Redundant Pattern Vector

After mining all the closed sequential patterns, we convert the input sequences to length-weighted pattern vectors. Specifically, a weight (W) is calculated as follow, which is motivated by [24]:(1)$$W=\mathrm{PatLength}\times \log_{2}\frac{\mathrm{DBSize}}{\mathrm{PatSup}}$$

In Formula (1), PatLength is the length of the current pattern. DBSize is the size of the input sequence set, and PatSup indicates the pattern absolute support (number of sequences contain such pattern). Intuitively, a pattern is less discriminative if it appears in many sequences. Therefore, we use a logarithm function to indicate the negative correlation between pattern support and its weight. We also consider the effect of pattern length by adding a multiplex operator to the weighting function.

### 2.4. Distance Matrix and Tree Construction

We apply the Jensen-Shannon (JS) distance metric [25] to measure distance between obtained pattern vectors. The JS distance between two vectors P and Q is defined as
(2)$$\mathrm{JS}(P,Q)=\frac{1}{2}\mathrm{KL}(P,M)+\frac{1}{2}\mathrm{KL}(Q,M)$$
where
(3)$$M=\frac{1}{2}\left(P+Q\right)$$

The KL(P,M) is the Kullback-Leibler divergence [26] between P and M defined as
(4)$$\mathrm{KL}(P,M)=\sum_{i}{P(i)\log}_{2}\frac{P(i)}{M(i)}$$

After calculating the distance between all pairs of sequences, we obtain a distance matrix. Then we use the Neighbor joining method as implemented in the software package PHYLIP [27] to reconstruct the phylogenetic tree.

### 2.5. Reference Tree and Tree Comparison

To quantitatively estimate the quality of phylogenetic trees reconstructed from the distance matrix calculated from pattern vectors, we compare trees constructed by different method toward the reference tree, a “true” history of simulated data or tree built by MEGA7 for BALIBASE dataset. The treedist implemented in PHYLIP was used to compute the distance between two trees. Additionally, we use symmetric difference [28], which is based on Robinson-Foulds distance to measure the topological differences between two trees. And intuitively the smaller the distance is, the more similar current tree is to the reference tree.

### 2.6. Code Availability

The source code is public available at https://github.com/xjtu-omics/TreeDM.

## 3. Results

For large amount of divergent sequences, using feature space to construct informative pattern vectors and build accurate phylogenetic tree from sequences is the focus of our study. In the pattern mining process, there are three parameters have to be tuned according to the characteristics of the input sequences. We test the effect of chosen different parameters on the results. The effects of non-wildcard/max-wildcard and support are discussed in Section 3.1 and Section 3.2, respectively. In Section 3.3 and Section 3.4, we compare the performance of various methods on simulated datasets with different sequence similarity and size. Furthermore we test the performance of different methods on the BALIBASE 3.0 dataset in Section 3.5. Finally, we compare the runtime of different methods.

### 3.1. Non-Wildcard and Max-Wildcard

In our specifically designed sequential pattern mining algorithm, we include two parameters, non-wildcard and max-wildcard to specify the minimum number of non-wildcard matches and the maximum number of consecutive wildcard allowed for each resulting pattern. As shown in Figure 2, we explore various settings for non-wildcard and max-wildcard on sequences of different similarities ranging from ~20% to ~5%, which are measured as relatedness from similar to divergent in Rose. We infer the similarity between sequences is approximately 20%/15%/10%/7%/5% under relatedness 250/300/350/400/450. We run Rose with default parameters except for the relatedness values and sequences selected from leaves. You can find examples of sequence sets (pamfile1-5) with different similarities in the Appendix A. Performance comparison is conducted on simulated sequences with different similarity.

Generally, as shown in Figure 2, the result is improved when we fix the number of non-wildcard, since patterns will contain more max-wildcards at such setting but result in more sequential patterns. However, calculating the distance matrix using high-dimensional pattern vectors is time consuming, especially for highly similarity sequences. For highly similarity sequences, we can construct phylogenetic tree using small number of patterns. Redundant patterns do not improve the result, and sometimes will lead to worse result. In order to get accurate result and faster speed, for divergent sequences, we set both non-wildcard and max-wildcard to be 2. There is a little difference between the quality of phylogenetic tree when non-wildcard is set to 2 or 3.

### 3.2. Support in Sequential Pattern Mining

In data mining, the term “Support” means the number of sequences contain a specific pattern. In general, as the support increases, the number of patterns used to construct the feature space decrease. As shown in Figure 3, for similar sequences (relatedness value is 250, similarity is about 20%), the distance between phylogenetic tree built by our approach and reference tree becomes smaller as the minimal support value increases. For divergent sequences (relatedness value is 450, similarity is about 5%), we observe opposite result. Thus, we recommend support should be set according to the sequence similarity.

### 3.3. Sequence Similarity and Performance

For highly similarity sequence sets, it is rather straightforward to obtain an accurate phylogenetic tree by both MSA approach and alignment free methods. It is, however, difficult to construct a tree when sequence is dissimilar. In order to examine the performance of different methods on different similarity data sets. We choose a set of 200 protein sequences of different relatedness with a length of about 350 aa that has been used in Section 3.1. An example of sequence sets (pamfile1) with different relatedness can be found in the Appendix A. Because when the sequence similarity is low, MSA fails to produce an accurate alignment. The existing methods we use for comparison are feature frequency profile (FFP), CVTree (CV), spaced word (SW), average common substring (ACS) and underlying approach (UA). For FFP, CV, SW methods, we use either l or k values for the best results for different data sets, and recommended values for some other non-primary parameters if available. We set the minimum length of substring in UA to 2 as recommended. In the sequential pattern mining process, we consider the weight of the pattern and set support parameter as 0.03. The values of non-wildcard and max-wildcard are 2 and 2, respectively. PVTree represents pattern vector and JS distance based phylogenetic tree construction approach. As shown in Figure 4, when the sequences become divergent, our method outperforms other methods.

### 3.4. Numbers of the Sequences and Performance

In addition to construct phylogenetic tree in low similarity sequences, we also examined our approach on large-scale data sets. Here we generate 200, 400, 600, 800, 1000 simulated sequences with 450 relatedness value and 350 aa length. You can find example of sequence sets with different numbers in the Appendix A. We set the parameters of methods for comparison in the same way as in the Section 3.3. As shown in Figure 5, our method consistently outperforms competing methods with various numbers of sequences for phylogeny construction.

### 3.5. BALIBASE Data Sets

To investigate performance of our method on real data, we compare with other methods using the BALIBASE3.0 (RV911-BOX270) with sequence similarity below 20%. The data of RV911-BOX270 can be found in the Appendix A. Since the similarity is not very low, we first construct phylogenetic tree via ClustalW [29] and Neighbor Joining method both from MEGA7. We run ClustalW with default parameters to obtain sequence alignment result. The protein weight matrix was set to BLOSUM. Then we use the Neighbor Joining method with the default parameters to build the phylogenetic tree. The reference tree was built by Maximum Likelihood method with the default parameters based on aligned sequences provided by BALIBASE. The distance between MSA based tree and reference tree is 64, which is the best. This suggests that the phylogenetic tree constructed based on MSA is better than alignment free based method when the sequence similarity is not very low. As shown in Figure 6, five alignment free methods for comparison, their parameters are selected in the same way as in Section 3.3. We consider the weight of the pattern and support is 0.05. The values of non-wildcard and max-wildcard are 2 and 1, respectively. Clearly, when the sequence similarity is below 20%, our method (PVTree) can get an ideal result.

### 3.6. Runtime

In order to compare the run time of our method and other alignment free approaches, we test on 200 simulated protein sequences of 350 aa used in Section 3.3. The relatedness is 450. As shown in Table 1, our method is much faster than UA and FFP but comparable with ACS, CV and SW. It is worth noting that the distance between phylogenetic tree built by our method and the reference tree is minimum.

## 4. Discussion

In this study, instead of using k-mer or substring based alignment free method, we propose sequential pattern mining based approach to identify patterns shared among sequences and use them to measure similarity between sequences for phylogeny reconstruction. Based on results obtained from 200 simulated protein sequences, PVTree is able to build high quality phylogenetic tree for sequences with similarity higher than relatedness score of 350. As for divergent sequences, we found that our approach plus a weighted pattern vector yields accurate phylogenetic tree. We have demonstrate our sequential pattern mining based phylogeny reconstruction is efficient and reliable. However, for sequence sets with higher similarity, users may get a disappointing result if they construct phylogenetic tree directly by using default pattern vectors which take into account the weight information rather than using binary pattern vectors. The current version of PVTree has two major limitations, on one hand users have to set parameters manually according to the characteristics of input sequences. For example, for large-scale sequence sets with highly similarity, when users set inappropriate small support and non-wildcard value, the constructed tree is not reliable because of too many useless patterns. On the other hand, PVTree has not been well solved for low similarity DNA/RNA sequence sets, thus restricting users to directly use low similarity DNA/RNA to build phylogenetic tree. To further improve PVTree, we will consider situation of overlapping patterns and the pattern length which could be used to prioritize longer patterns. Compare with methods based on MSA, which can’t give a reliable alignment result for large scale low similarity sequence sets, and then can’t construct a reasonable phylogenetic tree, as well as the shortcomings of the methods based on k-mer or substring in pattern extraction or sequence similarity calculation. For low similarity sequence sets, a reasonable phylogenetic tree can be constructed based on sequential pattern mining algorithm.

The most important parameter in our approach is the support value. For datasets with different levels of similarity and number of sequences, the minimum support should be sufficient to identify patterns shared by two or more sequences. Only patterns shared by two or more sequences contain valuable information for phylogeny construction. However k-mer method does not select patterns shared among sequences but only uses a sliding window to extract fixed-length patterns regardless of their support in the dataset. Thus the resulting frequency vector is sparse and slow down the calculation. When the number of input sequences is large, a small support input parameter leads to large number of patterns shared by sequences and reduce the speed of the phylogenetic tree construction. Support shall be adjusted for datasets with different similarities. For divergent sequences, smaller support parameter yields more rare patterns, allowing construction of a high quality phylogenetic tree. However for the highly similar datasets, low support value will lead to excessive amount of patterns, slowing calculation as well as reducing the quality of the phylogenetic trees.

Currently we are using exact matches during the mining process. This means that the two patterns DRY and ERY are treated as two distinct patterns. Since residues D and E are comparable, i.e., both are negatively charged, one may combine them as one pattern (D/E)RY. One of the solutions to solve this is to define similar residues as equivalent so that we can combine their instances during mining. Another solution is to include a standard amino acid substitution matrix such as BLOSUM62 into sequential pattern mining. Thus similar patterns will be combined if their scores to the center pattern are above a given cutoff.

In future work, we can also mine patterns consider their order in the sequence, because if we consider motifs as the words, the sequential order of these words may also carry essential biological meaning. Sequences must also have their patterns in the correct order to fold correctly and function properly. We need compare them with normal patterns in phylogeny reconstruction. For larger sequence sets, inspired by the application of parallel computing in phylogenetic tree construction [30,31], we will also try to use parallel computing platforms to speed up phylogenetic tree construction.

In conclusion, alignment free method based on sequential pattern mining will be an alternative solution of phylogeny construction for deviating sequences.

## Figures and Tables

**Figure 1 genes-10-00073-f001:**
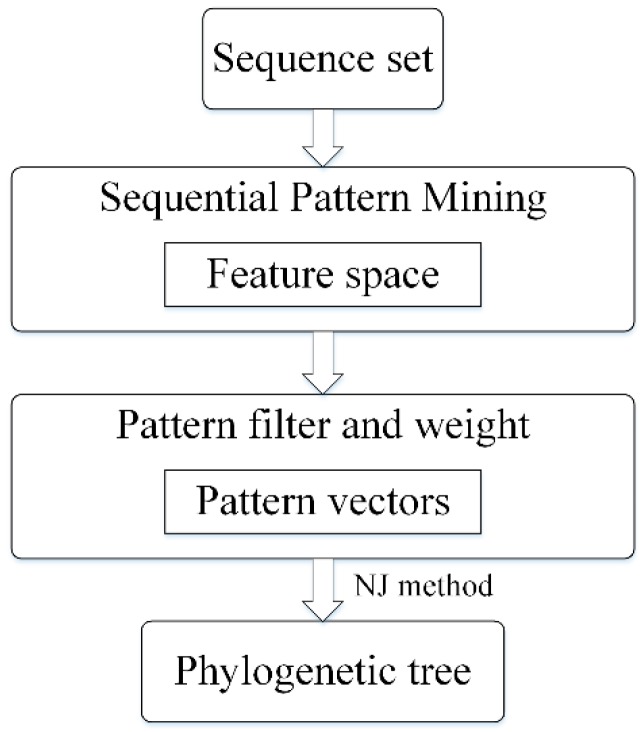
The flow chart for phylogeny reconstruction based on sequential pattern mining method. NJ: Neighbor Joining

**Figure 2 genes-10-00073-f002:**
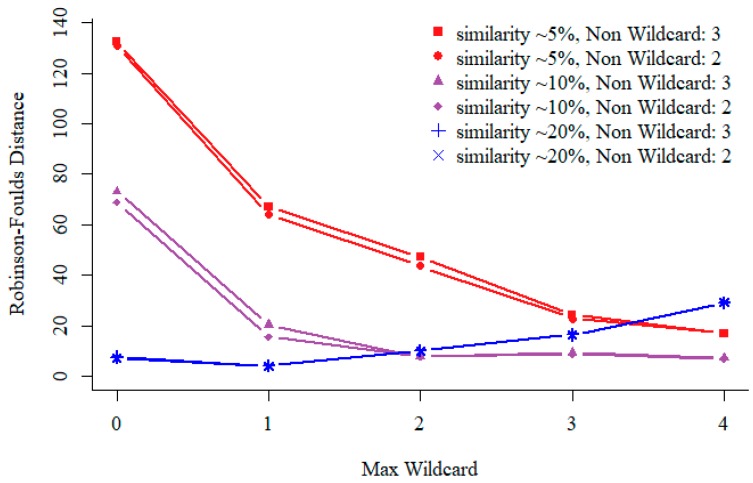
Performance comparison with different non-wildcard and max-wildcard settings on sequences of different similarities. Test results on 200 simulated protein sequences of length 350 amino acid (aa). For different relatedness sequences, all support of the sequential pattern mining is 0.03. For a fixed non-wildcard, we generated patterns of max-wildcard between 0 and 4, i.e., with 2 non-wildcard and up to 4 max-wildcards. The vertical coordinate represents the distance of phylogenetic trees.

**Figure 3 genes-10-00073-f003:**
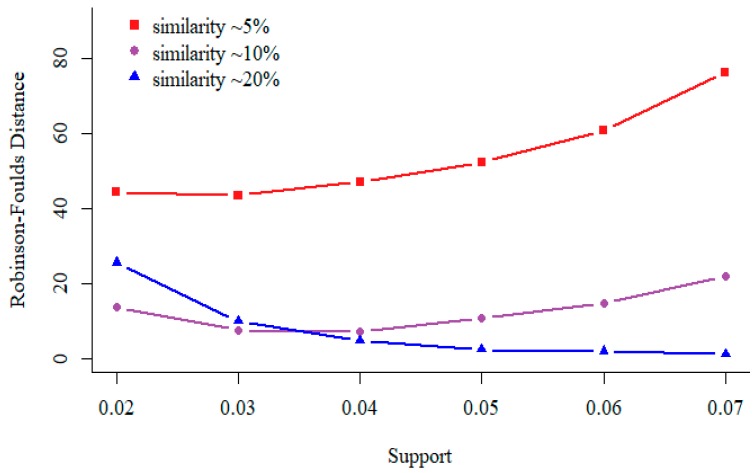
Test results on 200 simulated protein sequences of length 350 aa (used in Section 3.1). For different relatedness sequences, the values of non-wildcard and max-wildcard are 2 and 2, respectively.

**Figure 4 genes-10-00073-f004:**
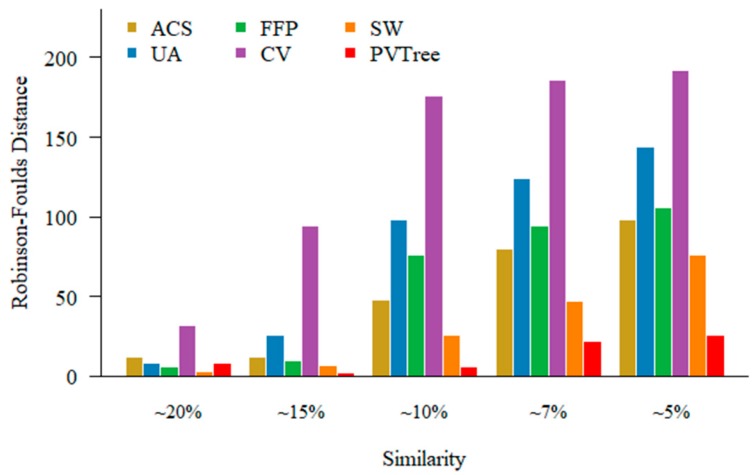
Performance of alignment free methods on different relatedness sequence sets. The methods we use for comparison are average common substring (ACS), underlying approach (UA), feature frequency profile (FFP), CVTree (CV), spaced word (SW).

**Figure 5 genes-10-00073-f005:**
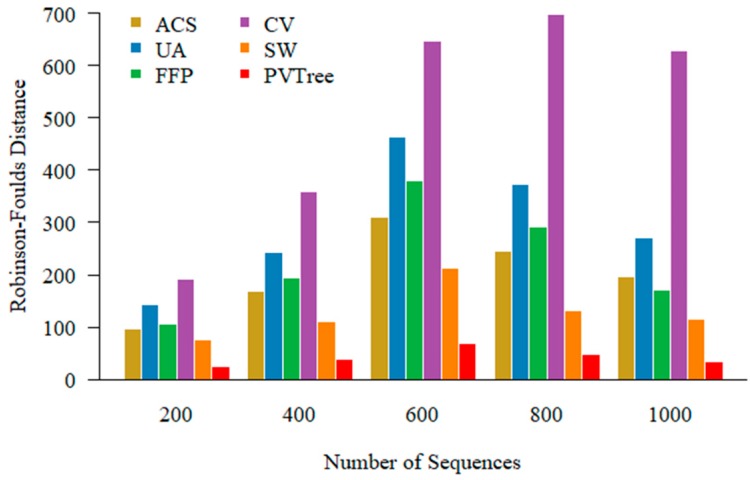
Tested on different number of sequence sets. The methods we use for comparison are average common substring (ACS), underlying approach (UA), feature frequency profile (FFP), CVTree (CV), spaced word (SW).

**Figure 6 genes-10-00073-f006:**
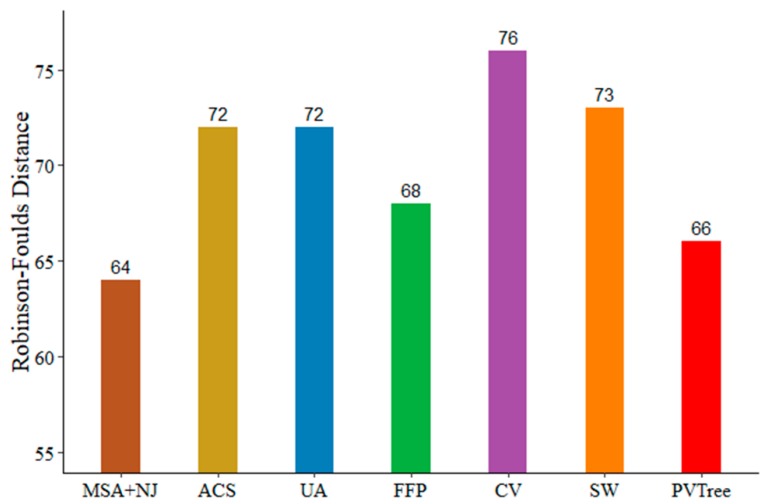
Test results on BALIBASE; PVTree method used with non-wildcard and max-wildcard are 2 and 1, respectively. The spaced word (SW) approach was used with a pattern weight of k = 4, feature frequency profile (FFP) was used with l = 6, CVTree (CV) was used with k = 9, and the minimum length of substring in underlying approach (UA) is 2 recommended by the author. It is worth noting that in order to clearly show the difference between the various methods, the starting value of the ordinate is 55.

**Table 1 genes-10-00073-t001:** Run time on 200 simulated protein sequences of length 350 aa.

Method	Runtimes (s)
ACS	1.459
UA	171.868
FFP (l = 5)	133.908
CV (k = 3)	3.455
SW (k = 4)	1.403
PVTree (2-1)	5.571
PVTree (2-2)	12.782

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
