# Peer review of "PVTree: A Sequential Pattern Mining Method for Alignment Independent Phylogeny Reconstruction"

_genes, 2019, doi:10.3390/genes10020073_

Round 1
Reviewer 1 Report
The authors proposed a novel phylogeny reconstruction method. The topic is important in the bioinformatics. The paper is well written and organized. English is easy to follow. The experiments could support the conclusion. However, I still have some suggestions to improve the manuscript quality.
I suggest the authors to supply a flow chart to show the main idea, which could benefit the readers for following.
The method could be compared with HPTree (http://lab.malab.cn/soft/HPtree/) and HAlign , which is the latest phylogeny reconstruction method.
Reviewer 2 Report
The article presents a new method of alignment-free phylogenetic reconstruction that aims to take into account global sequence features. The idea is to avoid useful information from being discarded compared to existing sliding-window-based alignment-independent methods. The authors do a good job of introducing the limitations of traditional, MSA-dependent phylogenetics, and of the shot kmer-based alternatives. However there needs to be clearer discussion of the limitations of the approach as well as its strengths. More comparison to other methods would be useful, along with greater detail on the datasets used.
Abstract:
The article would benefit from checking the language and grammar throughout. Although most are minor, a few obscure the meaning of the text. Example from abstract:
currently: "...difficult for large amount of deviating sequences."
replace with: "...difficult for large numbers of highly divergent sequences"
or "...difficult for large sequence sets with low sequence identity"
It might be worth being specific that the article will be discussing phylogenies of gene families rather than attempting to reconstruct the phylogenies of species.
Introduction:
When discussing deviating sequences, it would be useful to give some examples of when this occurs (e.g. fast evolving proteins under high selection pressure, robust cysteine-rich proteins that are highly tolerant of mutation, ancient protein superfamilies that have had long times to diverge).
Methods:
The methods are overall clearly written. The GitHub folder is clearly organised and the readme is sufficient.
It could be useful to include in the supplementary data a worked example for a small dataset so that e.g. the weighted pattern vectors can be seen.
Results:
When defining the similarity of a dataset, is sequence identity being used, or are similar, but non-identical residues counted as partial matches?
Is there any way to implement the equivalent of a bootstrap method to check tree robustness to data perturbation?
Robinson-Foulds distances are useful measures of overall topological difference. It would be useful for at least one example to indicate which regions of the tree differ. are differences randomly distributed throughout or are certain areas of the tree or certain sequences more likely differ between the methods?
Fig 3: Definitions of the abbreviations are scattered throughout the results. It would be good to have them all repeated in the figure legend.
Fig 3 and 4: Both figures may be easier to read if presented as further line diagrams.
There are comparisons made to FFP and CV methods. I would also like to see comparison to simple MSA-based NJ tree to corroborate the results in figure 5.
Balibase contains several example datasets.
Why was RV911-BOX270 used in particular?
Was it chosen at random or for particular features?
How many sequences does it include and how long are they?
Could additional datasets be put through the same pipeline to test whether the results are reproduced?
Figure 5: When presenting a graph with a Y axis that doesn't start at 0, it must be clearly indicated (especially when all other graphs start at y=0.
Could a be compared to a pairwise sequence identity (eg with sequence identity distance on the x axis, and sequence pattern vector distance on the y axis)?
Discussion:
It would be good to see an example or discussion
The limitations and possible errors of the approach need to be clearly stated (especially since the limitations of MSA-dependent phylogenetics is explained in the introduction). Like kmer approaches, the underlying evolutionary model is less explicit than in ML or bayesian phylogenetics and is more similar to neighbour-joining phylogenetics.
What situations could lead to erroneous results?
What situations could lead to correct results that an MSA-dependent phylogeny would get wrong?
Since the technique is presented as not just a speed increase, When would a technique like this be used as opposed to others?
How does it handle highly repetitious sequences?
Is there a difference between low similarity due to frequent substitution versus low similarity due to frequent insertion/deletions?
Round 2
Reviewer 2 Report
I commend the authors for their thorough response to both my and the other reviewer's suggestions. They have satisfied all of my questions and I will be happy to see the work published.